# Identifying the Common Cell-Free DNA Biomarkers across Seven Major Cancer Types

**DOI:** 10.3390/biology12070934

**Published:** 2023-06-29

**Authors:** Mingyu Luo, Yining Liu, Min Zhao

**Affiliations:** 1School of Science, Technology and Engineering, University of the Sunshine Coast, Sippy Downs, QLD 4558, Australia; mingyu.luo@research.usc.edu.au; 2The School of Public Health, Institute for Chemical Carcinogenesis, Guangzhou Medical University, Guangzhou 510120, China; yiningliu.pku@gmail.com

**Keywords:** early cancer diagnosis, cell-free DNA, biomarker, integrative biology, pan cancer

## Abstract

**Simple Summary:**

Blood-based circulating cell-free DNA (cfDNA) biomarkers are important for cancer detection because they can provide a less invasive and more cost-effective way of detecting cancer and offer the possibility of screening large populations at risk for the early detection of multiple cancers. However, highly sensitive techniques are needed to detect circulating tumour DNA (ctDNA), and further optimization and standardization of pre-analytical and analytical steps are required to harness the full potential of cfDNA analysis.

**Abstract:**

Blood-based detection of circulating cell-free DNA (cfDNA) is a non-invasive and easily accessible method for early cancer detection. Despite the extensive utility of cfDNA, there are still many challenges to developing clinical biomarkers. For example, cfDNA with genetic alterations often composes a small portion of the DNA circulating in plasma, which can be confounded by cfDNA contributed by normal cells. Therefore, filtering out the potential false-positive cfDNA mutations from healthy populations will be important for cancer-based biomarkers. Additionally, many low-frequency genetic alterations are easily overlooked in a small number of cfDNA-based cancer tests. We hypothesize that the combination of diverse types of cancer studies on cfDNA will provide us with a new perspective on the identification of low-frequency genetic variants across cancer types for promoting early diagnosis. By building a standardized computational pipeline for 1358 cfDNA samples across seven cancer types, we prioritized 129 shard genetic variants in the major cancer types. Further functional analysis of the 129 variants found that they are mainly enriched in ribosome pathways such as cotranslational protein targeting the membrane, some of which are tumour suppressors, oncogenes, and genes related to cancer initiation. In summary, our integrative analysis revealed the important roles of ribosome proteins as common biomarkers in early cancer diagnosis.

## 1. Introduction

Cancer results in the release of cell-free DNA (cfDNA) into the bloodstream through processes such as apoptosis and necrosis. As a result, cancer patients often exhibit elevated levels of cfDNA. This phenomenon has sparked significant interest in utilizing circulating cfDNA as a “liquid biopsy” for noninvasive early detection of cancer [1]. In general, cfDNA with genetic alterations constitutes a small proportion of the DNA circulating in plasma, which can be confused with cfDNA from normal cells [1]. Therefore, it will be essential for cancer-based biomarkers to eliminate potential false-positive cfDNA mutations in the healthy population. In addition, many low-frequency genetic alterations are easily missed in cancer tests based on a small amount of cfDNA.

Cancer genome project advancements and new applications of next-generation sequencing (NGS) technology have facilitated ground-breaking research on cfDNA over the past decade. However, the development of clinical biomarkers continues to face significant obstacles. Firstly, the sensitivity of current cfDNA profiling strategies is insufficient for the simultaneous detection of multiple cancers. This can be enhanced in several ways, including by optimizing the pre-analytical steps, collecting samples from body fluids with higher mutation allele fractions, and enriching tumour-derived cfDNA after extraction. By combining multiple biomarkers into a single evaluation, the sensitivity and specificity of cfDNA tests can be dramatically improved. Secondly, the quantitative and qualitative fluctuations of cfDNA in a person’s blood impede the reproducibility of measurements, interpretations, and comparisons. A better understanding of the cfDNA release rate could help solve this issue [2]. Thirdly, it is necessary to validate the quantification of cfDNA, subsequent mutation analysis, and other analytical steps, including the sequencing platform itself, in order to simulate the clinical environment [3]. Lastly, the use of diverse high-throughput sequencing platforms makes it difficult to reproduce results, highlighting the need for standardization and analytical validation of liquid biopsy techniques [4].

In a healthy individual, mutations in oncogenes and tumour suppressor genes play an important role in the beginning stages of cancer [5,6]. There are fewer than 2000 genes, despite the fact that these important driver genes are essential for cancer diagnosis. On the other hand, cancer cells typically contain thousands of mutations that do not directly drive cancer initiation and progression, and these mutations can also be found in healthy populations. In this study, we collected data on 1358 cfDNA-based samples with original sequences from 14 different projects in order to focus on key events in the progression of cancer. These experiments involve seven different types of major cancers, including head and neck cancer, lung cancer, breast cancer, prostate cancer, gastric cancer, colon cancer, and liver cancer. We hypothesize that the combination of diverse types of cancer studies on cfDNA will allow us to identify high-quality genetic variants across cancer types for early clinical cancer detection [5].

## 2. Materials and Methods

### 2.1. Data Sources and the Data Filtering Pipeline

As shown in Figure 1A, our analysis pipeline was started by downloading data from the NCBI SRA database (http://www.ncbi.nlm.nih.gov/sra, accessed on 10 April 2019) using the SRA Toolkit (https://www.ncbi.nlm.nih.gov/sra/docs/toolkitsoft/, accessed on 10 April 2019). The SRA database is a public repository for millions of publicly available data related to genomic sequencing. For our project, we only focused on the cell-free DNA data in cancers. Therefore, we searched the SRA database by the following expressions: “cell free DNA [title] or single cell DNA [title] or single cell RNA [title]” and “cancer or tumour” on 10 April 2019. Then we downloaded 1358 samples with raw sequences from a total of fourteen projects involving seven major cancer types, including breast, colorectal, head and neck, liver, lung, prostate, and stomach cancers.

### 2.2. Sequence Data Alignment and Pre-Processing

The raw data downloaded from the SRA database are short reads in the Fastq format. To translate the raw data into meaningful information, we adopted the best practices of the Genome Analysis Toolkit (GATK) for the overall data pre-processing and genome mapping. Firstly, we aligned the raw Fastq reads to the Human reference assembly HG19 by using the genome aligner BWA v0.7.13 [7] with default settings. The resulting binary alignment map (BAM) files were used as input for the tools used in the GATK best practice [8]. In brief, we removed the potential duplicated short reads with Picard’s MarkDuplicates command. We also corrected the local alignment around indels based on GATK’s Indel-Realigner module. The recalibration of the quality score and reduction of machine-read error was further conducted by using GATK’s base quality score recalibration (BQSR) module (Figure 1A).

### 2.3. Variant Calling, Filtration, and Annotation

The pre-processed BAM files with recalibrated quality scores were further analysed by the somatic mutation calling tools of MuTect2 [9] and Monovar [10] for single-cell DNA data. In detail, the variant calling format (VCF) files were generated from two variant calling tools for each sample. Then the VCF files were used as input to eliminate potential sequencing and germline artefacts. For example, we removed those non-functional variants and focused on those somatic variants detected in three or more cancer types. In addition, we also removed the false-positive genetic mutations that may be present in the VCFs (Figure 1B). Then, functional annotations for variants were added to each mutation using the ANNOVAR software v. 24 October 2019 [11]. In addition, the pathogenicity of missense variants was predicted in silico using scores from dbNSFP [12] based on 12 different algorithms, such as SIFT and CADD [13].

### 2.4. High-Quality Variants Prediction

It is crucial to accurately predict the deleteriousness of nonsynonymous variants in order to distinguish pathogenic mutations from background polymorphisms [13]. Although numerous methods for predicting deleteriousness have been developed, their prediction results are sometimes inconsistent [14]. The computational algorithms utilised by these prediction methods (Markov model, evolutionary conservation, random forest, neural network, etc.) vary. Therefore, it is recommended to use multiple prediction algorithms for variant evaluation to eliminate algorithm bias [15]. We chose Combined Annotation-Dependent Depletion (CADD) [16] and Functional Analysis through Hidden Markov Models with an eXtended Feature Set (FATHMM-XF) [17] as our prediction algorithms based on their relative merits. In brief, CADD assesses the deleterious nature of SNVs based on a variety of genomic characteristics, including the surrounding sequence context, epigenetic measurements, evolutionary constraints, and functional predictions [16]. CADD’s ability to prioritise functional, deleterious, and pathogenic variants is unmatched by any single-annotation method currently in use [18]. Compared to traditional procedures (such as SIFT), CADD was determined to be the most effective in silico algorithm in previous SNV pathogenicity analyses [19]. However, the disadvantage of CADD is limited accuracy for predicting variants in non-coding regions [20]. To add non-coding information, we used FATHMM-XF, one of the most efficient tools for non-coding regions [17].

### 2.5. Functional and Pathway Enrichment Analysis

To investigate the functional patterns of the genes associated with the identified somatic mutations, we conducted a comprehensive functional annotation. In brief, significant gene ontology (GO) biological process terms and Kyoto Encyclopedia of Genes and Genomes (KEGG) pathway enrichment analyses were performed to analyse the identified biomarkers at the functional level. GO provides a general framework to characterise the gene functions shared in multiple species [21]. According to the adjusted statistical *p* values, the terms were arranged in ascending order, making it simple to focus on the most significant GO terms associated with the biomarker genes. To supplement the missing information in the GO annotation, we also consulted the KEGG database for pathway information. KEGG assigns specific gene set pathways to key data containing higher-order functional information and can be used for the functional interpretation and practical application of genomic data [22]. In practice, all human genes as the background and the identified biomarkers as the input were used to perform GO function and KEGG pathway enrichment analysis, and FDR 0.05 was considered statistically significant using Toppfun [23].

### 2.6. Protein-Protein Interaction and Hub Gene Analysis

To understand the metabolic and molecular mechanisms related to the identified biomarkers shared in multiple cancers, we utilised the existing protein-protein interaction data. In brief, the Search Tool for the Retrieval of Interacting Genes (STRING) database (version 10.0) [24] provides a comprehensive analysis and integration of protein-protein interactions, including direct physical connections and indirect functional associations such as co-expression in multiple datasets. The output from the STRING results was further visualised by using Cytoscape 3.7.1, which makes it easy to depict the genes from different functional groups [25]. In addition, the plug-in app cytohubba in Cytoscape was downloaded and installed to explore the hub genes [26]. Using the top scores of the Maximal Clique Centrality (MCC) algorithm, the hub genes with high connectivity in the gene expression network were eliminated and clustered.

### 2.7. Survival and Mutational Analysis of the Top Module Genes in the TCGA Database

Using data from 10,953 patients from 33 TCGA pan-cancer studies, we further explored the potential clinical application of those key genes identified in the network modules. For instance, mutational analysis was performed to investigate the single-nucleotide somatic mutation and copy number variation patterns of the genes from the top module at a pan-cancer level [27]. The frequency of genetic alteration was further plotted based on the number of tumour samples containing the somatic mutation and copy number alteration associated with the key network genes. Additionally, we associate the genes with patient overall survival data from TCGA by classifying all patients into altered and unaltered groups using cBioportal [28]. To focus on reliable results, the log-rank analysis and Kaplan–Meier plots were generated.

## 3. Results

### 3.1. Identification of Potential Biomarkers in cfDNA

To collect the high-quality genetic variations in cfDNA for liquid biopsy biomarkers, we searched SRA and downloaded raw sequence data from 14 projects involving seven major cancer types. Firstly, we performed a gene-based annotation of all called variants to remove non-functional variants and identified a total of 896,193 exonic SNVs or indels (Figure 2A). Secondly, to further minimise the rate of false-positive calls, variants from different cancer types were combined and duplicate variants were removed, leaving a total of 858,176 variants. Figure 2B,C show how the variants were distributed and shared across different cancer types. Thirdly, variants present in at least three cancer types were selected to refine the list to 6981 for downstream analysis. A total of 129 variants were predicted to be deleterious by a combination of two pan-genome prediction scores (CADD and FATHMM-XF). The 116 corresponding potential biomarkers were then used for further analysis (Appendix A).

### 3.2. KEGG Pathway Analysis Confirmed the Close Relationship between Ribosome and Cancer

As shown in Figure 3A, the top GO terms of cellular component, molecular function and biological process include cotranslational protein targeting to the membrane (adjusted *p*-value = 1.326 × 10^−21^), protein targeting to ER (adjusted *p*-value = 2.894 × 10^−19^), translational initiation (adjusted *p*-value = 8.101 × 10^−17^), mRNA catabolic process (adjusted *p*-value = 2.941 × 10^−16^), the establishment of protein localization to the endoplasmic reticulum (adjusted *p*-value = 5.164 × 10^−19^), and cytosolic ribosome (adjusted *p*-value = 4.468 × 10^−20^) (Appendix A).

Additional signalling pathway analyses (Appendix A) were conducted. For instance, the KEGG analysis showed that the biomarkers were mainly enriched in the ribosome, oxidative phosphorylation, proteasome, and other signalling pathways (Figure 3B). Ribosomes, for instance, are important for the translation of mRNA-contained information into functional proteins, which align well with the enriched GO function “cotranslational protein targeting to membrane” [29]. More interestingly, hyperactivation of ribosome biogenesis, which can be triggered by oncogenes or the loss of tumour suppressor genes, plays an essential role in the initiation and progression of cancer [30]. Recent studies suggest that both increased numbers and altered modifications of ribosomes may contribute to cancer development. For instance, multiple cancers, including endometrial cancer, high-grade gliomas, colorectal cancer, acute and chronic lymphocytic leukaemia, have been found to contain ribosomal genetic mutations [31].

The OXPHOS (oxidative phosphorylation) metabolic pathway is another significant pathway that deserves a mention. Among the 116 biomarker genes identified in this study, 8 are from the OXPHOS pathway, including NDUFC2-KCTD14, ATP6V0B, COX4I1, COX7A2, NDUFA3, UQCR10, NDUFB10, and NDUFC2. These genes are shared by breast, liver, and lung cancers, based on our data. It produces ATP by transporting electrons to the electron transport chain, a series of transmembrane protein complexes in the mitochondrial inner membrane (ETC) [32,33]. Cancer cells require OXPHOS, and cancer stem cells are frequently characterised by an increased reliance on OXPHOS [34]. Many studies have assumed that OXPHOS is downregulated in all cancers because cancer cells have a higher glycolysis rate than normal cells [34]. Additionally, the downregulation of OXPHOS is frequently correlated with poor clinical outcomes and metastasis [35]. Inhibition of OXPHOS has also been shown to reduce oxygen consumption rate (OCR) and alleviate hypoxia in tumours [32].

### 3.3. Network Analysis Revealed Hub Genes Associated with Cancer Development

To evaluate the interactive relationships among the identified biomarkers, we mapped them to the STRING database. The final interactome contains 115 genes and 477 connections. In the network, the average node degree is 8.3, and the average local clustering coefficient is 0.506. The Protein-protein interaction (PPI) enrichment *p* < 1.0 × 10^−16^ (Figure 4A). In summary, these topological characteristics of the network indicate that the genes within it can exchange information efficiently.

Then, we utilised the Molecular COmplex DEtection (MCODE) application to identify clustered modules throughout the entire network. The network consisted of 5 modules, with the top module containing 25 nodes and 262 edges. The 25 genes in the top module were selected for alteration frequency and survival analysis (Figure 4B). Based on the MCC (Maximal Clique Centrality) scores, we prioritise the most stable hub genes in the network, including RPS15A, RPS23, RPS9, RPS21, RPS14, RPS25, RPS6, RPL27, RPL35A, and UBA52 (Figure 4C). Among these genes, RPS15A (Ribosomal protein s15a) was shown to be related to many cancers in previous studies. As a component of the 40S subunit, increased RPS15A expression is closely correlated with poor prognosis in gastric cancer (GC) patients and promotes epithelial-mesenchymal transition (EMT) and GC progression, as demonstrated [36].

### 3.4. Overlapping with OCGs, TSGs, and CIGs Revealed Multiple Roles Played by Identified Biomarkers

In order to evaluate the roles of the potential biomarkers in cancer progression, we mapped the genes to known oncogenes (OCGs) [6], tumour suppressor genes (TSGs) [5], and cancer initiation genes (CIGs) [37]. This analysis identified 10 biomarker genes reported as either CIGs, OCGs, or TSGs (Figure 4D) (Appendix A). These genes included DUSP12, VIM, FOS, UBE2C, MIEN1, HINT1, LITAF, GABARAP, PFN1, and MLF2. As a member of the E2 ubiquitin-conjugating enzyme family, UBE2C is overexpressed in all 27 cancers, and patients with higher UBE2C expression levels exhibited a shorter overall survival duration [38]. Another interesting gene is LITAF (Lipopolysaccharide-induced tumour necrosis factor-α factor). It possesses transcription factor activity and is involved in the regulation of protein quality. Previous research has suggested that LITAF functions as a TSG and is frequently underrepresented in the prostate, pancreatic, and stomach cancers [39]. Taken together, these findings confirmed the significance of these biomarkers in the development of cancer, indicating their potential use in clinical diagnosis.

### 3.5. Patients with Altered Genes in the Top Functional Module Have a Significantly Worse Overall Survival Rate

The frequencies of genetic alterations in the 25 genes in the top module were evaluated using the cBioPortal database. Approximately 27% of clinical cases from 32 different cancer studies exhibited significant alterations in the 25 genes (Figure 5A). Kaplan-Meier plots were used to compare Overall survival in 10,953 patients with or without alterations in the 25 hub genes (Figure 5B). It was revealed that cases with altered genes exhibited significantly worse OS compared to those with unaltered genes (*p* value = 6.639 × 10^−5^).

## 4. Discussion

Numerous studies have demonstrated the potential of cfDNA as a biomarker for the early detection of cancer. However, the accuracy of cfDNA-based tests faces significant obstacles [2]. Previous cfDNA studies focused on a single tumour type or the results from a single cohort study [40,41], but there is no systematic examination of high-quality variants in different cancer types. In order to collect high-quality biomarkers in cfDNA, we constructed a computational pipeline to screen genetic variants shared by multiple tumour types based on the raw sequence data in the SRA databases.

In total, we identified 116 potential biomarkers following variant calling and filtering pipeline. As was suggested by functional enrichment analysis, these biomarker genes were mainly involved in the ribosome pathway, confirming the close relationship between the ribosome and cancer development, which contradicts the view held until recently that ribosomes played a rather passive role as the only molecular factory in the translation process [42]. Recent studies have linked the altered ribosome and dysregulated expression of specific ribosomal proteins to cancer initiation, evolution, and progression (RP) [43]. As an example, the correlation between accelerated colorectal cancer (CRC) cell growth and alterations in particular steps of ribosome biogenesis is cited as a key factor in cancer initiation [44]. Erica Buoso et al. provided an analysis of how ribosomes translate cancer progression in breast cancer through the ribosomal protein RACK1 [45]. Amandine et al. provided evidence supporting the role of altered ribosome components in the development of cancer and argued that ribosomes may play a crucial role in the acquisition and maintenance of the cancer stem cell phenotype [42]. Our study confirmed the association between ribosomes and cancer through the statistical analysis of large-scale genomic data from multiple cancer types. It also indicated that targeting the ribosome pathway is another promising possibility for developing a cancer therapeutic strategy.

The Circulating tumour DNA (ctDNA), which is a portion of the cfDNA released from the blood of cancer patients by tumour cells via apoptosis, necrosis, or active release, is another intriguing aspect of our data. As a new type of cancer biomarker, tumour-specific mutations in the ctDNA sequence can be used to identify cancer patients. To evaluate tumour heterogeneity, cfDNA-based liquid biopsy is less invasive, more feasible, and more comprehensive than tissue biopsy due to the rapid development of next-generation sequencing (NGS) technology. However, the use of ctDNA sequencing for cancer screening and early diagnosis is hindered by a low concentration of ctDNA in the blood and an increase in false positives resulting from normal healthy cells. This study developed a systematic pipeline that integrated a combination of prediction algorithms with optimised parameters to analyse raw sequencing data of cfDNA from various cancer types and identify high-quality variants in order to identify reliable biomarkers for cfDNA tests.

In general, ctDNA is released into the bloodstream by tumour cells or other components of the tumour microenvironment, such as cancer-associated fibroblasts (CAFs) or immune cells. In brief, ctDNA released from tumour cells or the tumour microenvironment can have effects on other tissues. For example, ctDNA can be taken up by immune cells, potentially modulating immune responses [46]. Additionally, the genetic alterations present in ctDNA may have implications for other tissues, potentially contributing to the development of secondary malignancies or affecting normal cellular function [47].

Tumour-derived ctDNA and normal cell-derived ctDNA can have different functions, primarily due to their distinct origins and genetic characteristics. Tumour-derived ctDNA contains genetic alterations that are specific to the tumour, such as oncogenic mutations or tumour suppressor gene alterations. These genetic changes can influence the behaviour of tumour cells, including their proliferation, survival, and response to therapy [46]. In addition, tumour-derived ctDNA can have direct effects on the tumour microenvironment and distant tissues [48]. It may contain information about the tumour’s biological characteristics, such as the presence of immune cell infiltration, angiogenesis, or stromal activation. Tumour-derived ctDNA can potentially modulate immune responses or contribute to the development of secondary malignancies [46,47]. Unlike tumour-specific ctDNA, normal cell-derived ctDNA contains genetic information representing normal cellular functions and does not harbour tumour-specific alterations. Normal cell-derived ctDNA, being derived from healthy cells, is less likely to exert the same effects on tissues as tumour-derived ctDNA.

Overall, tumour-derived ctDNA has significant clinical applications in personalised medicine [48]. By analysing the genetic alterations present in tumour-derived ctDNA, clinicians can identify specific targets for therapy, assess the potential for treatment resistance, and monitor the emergence of genetic changes associated with metastasis or disease recurrence. Normal cell-derived ctDNA, although less specific to cancer, may have other clinical applications, such as monitoring overall health or assessing the presence of non-cancerous genetic abnormalities.

Raw sequence data in FASTQ format were downloaded from the publicly available SRA database; these data come from 14 projects involving seven major cancer types. By applying the systematic pipeline, 116 biomarker genes shared by different cancer types were screened out from a total of 896,193 exonic SNVs or indels. Functional enrichment analysis shows that these biomarker genes are mainly involved in the ribosomal pathway, implying a close relationship between ribosomes and cancer development. By cross-referencing these 116 biomarker genes with known oncogenes, tumour suppressor genes, and cancer initiation genes, 10 genes were identified with multiple roles in cancer development. Then the importance of these biomarkers in cancer development was confirmed, implying their potential application for clinical diagnosis. In summary, this study provided new insight into identifying high-quality genetic variants in cfDNA across different cancer types, enabling a better application of cfDNA as a non-invasive diagnostic clinical biomarker for the early detection of cancer.

## 5. Conclusions

In this study, we developed a computational pipeline to identify high-quality biomarkers in cfDNA by screening genetic variants shared by multiple tumour types. Using a standard computational pipeline and 1358 cfDNA samples from seven cancer types, we ranked 129 shard genetic variants in seven major cancer subtypes. The majority of the 129 variants were enriched in ribosome pathways such as co-translational protein targeting and oxidative phosphorylation, which are associated with tumour suppressors, oncogenes, and cancer-initiating genes. Our integrative analysis revealed that ribosome proteins and oxidative phosphorylation enzymes are common cancer biomarkers.

## Figures and Tables

**Figure 1 biology-12-00934-f001:**
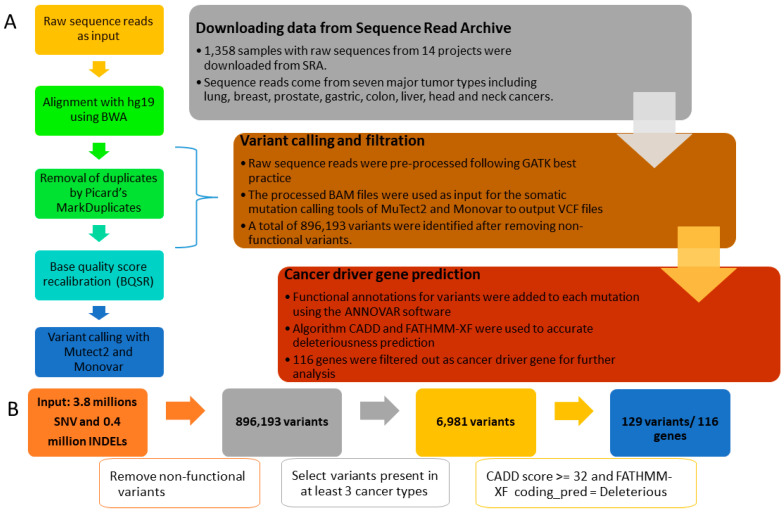
Flow chart of cfDNA data processing and filtering. (**A**) A three-stage workflow to identify the cancer driver genes in cfDNA sequence read, and five detailed steps for variant calling process on the left. (**B**) Flowchart for variants filtering, annotation, and deleterious gene prediction.

**Figure 2 biology-12-00934-f002:**
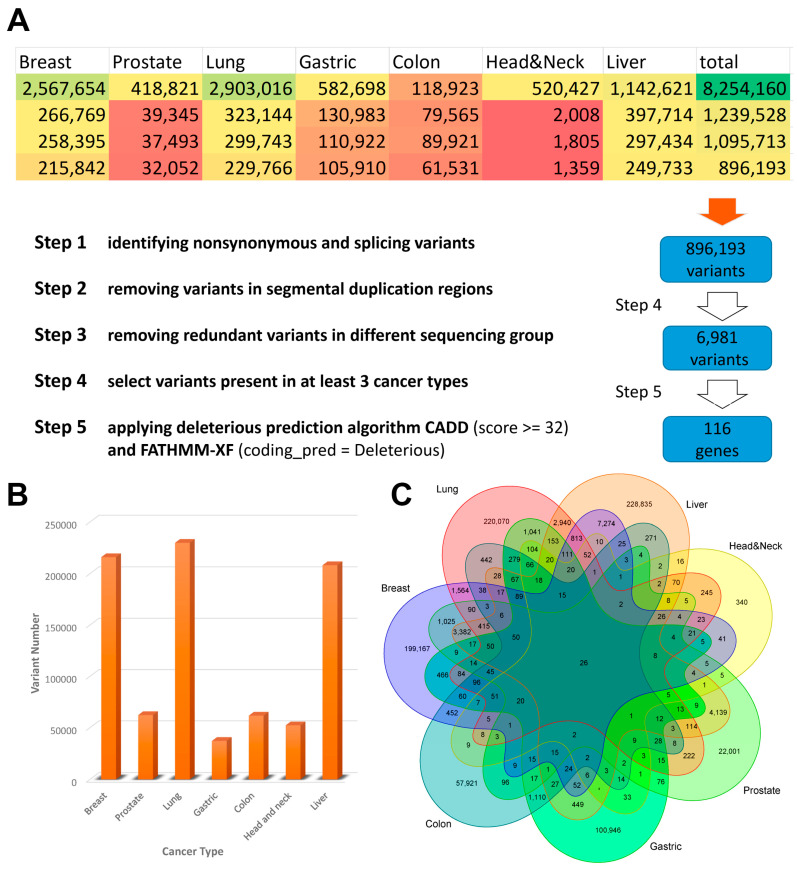
Summary of cfDNA sequence data processing results. (**A**) Detailed steps for variant calling, filtration, and biomarkers prediction. (**B**) A bar chart indicates the number of variants called for different cancer types. (**C**) Venn diagram depicts the overlap of somatic variants detected in various cancer types.

**Figure 3 biology-12-00934-f003:**
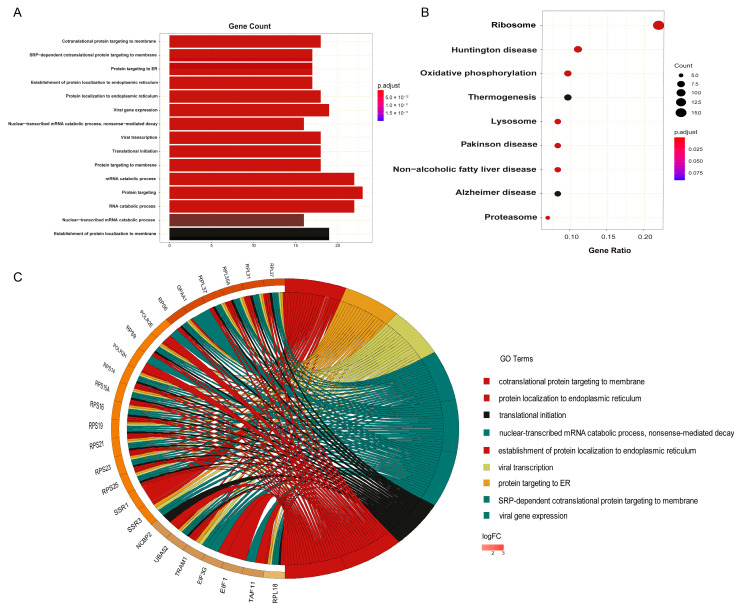
The functional enrichment analysis of potential cancer biomarkers in cfDNA. (**A**) The significantly enriched gene ontology (GO) terms. (**B**) All enriched KEGG pathways with statistical significance. (**C**) The overlapping of the top GO terms and the most frequent mutated genes.

**Figure 4 biology-12-00934-f004:**
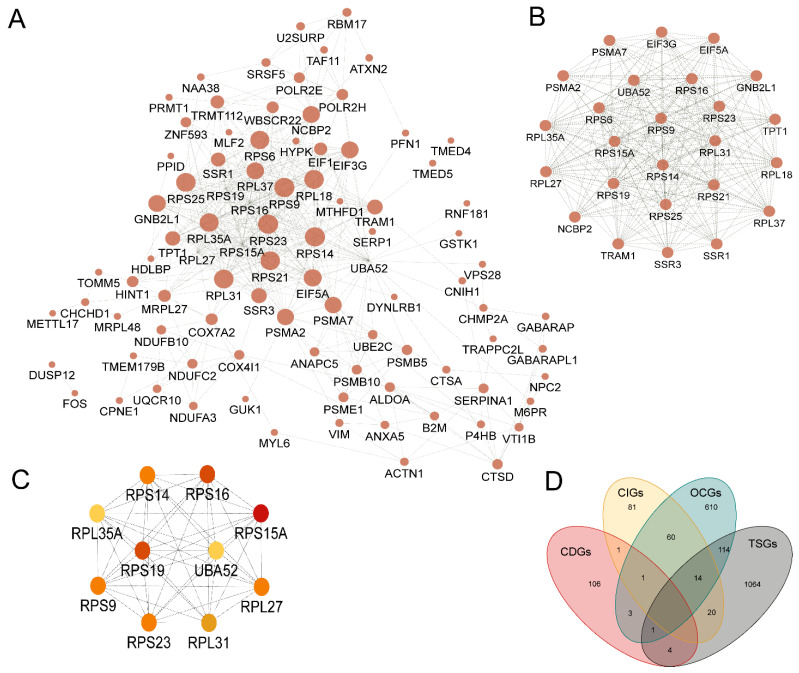
The network analysis of potential cancer biomarkers in cfDNA. (**A**) Visualized PPI analysis of biomarker genes. (**B**) 25 genes in module 1 with the highest Maximal Clique Centrality (MCC) scores. (**C**) Interconnection of 10 hub genes; the colour represents MCC scores, darker is higher. (**D**) The overlap of identified biomarkers (CDGs) with CIGs (cancer initiation genes), OCGs (oncogenes), and TSGs (tumour suppressor genes).

**Figure 5 biology-12-00934-f005:**
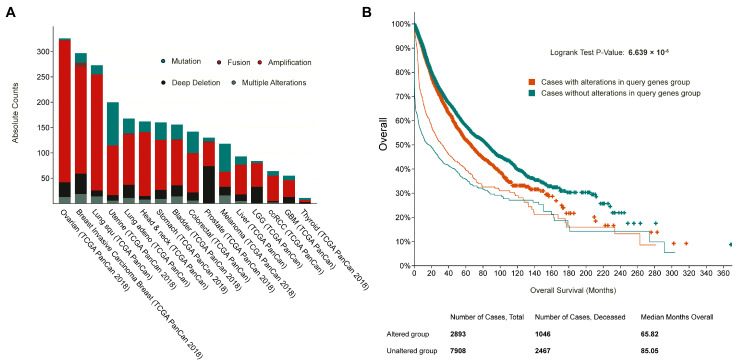
The mutational and clinical feature of the 25 genes in the top functional module. (**A**) The mutational frequency across multiple cancer types; the percentages of cases with the 25 altered genes were depicted in the y-axis. (**B**) The overall survival analysis of patients with altered (red) and unaltered (blue) 25 genes from the top functional module.

## Data Availability

The raw data used were from public Sequence Read Archive (SRA) databases.

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
