# Peer review of "Identifying the Common Cell-Free DNA Biomarkers across Seven Major Cancer Types"

_biology, 2023, doi:10.3390/biology12070934_

Round 1

Reviewer 1 Report

Comments:

1.       In the line 211 authors started discuss about oxidative phosphorylation but did not conclude the relation with common cell-free DNA biomarkers in different cancer types.

2.       Mention the full form of ctDNA in the manuscript.

3.       The ctDNA copy number in tumors is often reduced, but does cfDNA still increase? And if so, what explanation can be given for this mechanism?

4.       It would be good to describe any data or previous reports on the effect of ctDNA released from tumor cells/ stroma, e.g. CAFs or immune cells, on tumors or other tissues.

5.       Does tumor-derived or normal cell-derived ctDNA have different functions? Also, it would be better to explain in detail the methods used to distinguish between them.

Author Response

We have addressed the comments point-to-point in the attached PDF file. 

Reviewer 2 Report

In this manuscript, the authors re-analyzed 1358 cell-free DNA samples from seven major cancer types. Combining CADD and FATHMM-XF as the prediction algorithms, the authors identified 129 variants within 116 biomarker genes that were present in at lease 3 cancer types. Functional analysis identified its regulation in ribosome, association with cancer development. Further survival analysis revealed its correlation with an overall worse survival rate. Overall, the logic flow of this manuscript is clear to me. There are several concerns that might weaken the manuscript:

1. In the Materials and Methods, the authors used "single cell DNA" or "single cell RNA" as the key words for searching. What is the purpose of single cell DNA/RNA? How is this related to cell-free DNA from blood samples? Also, I didn't see any analysis about RNA expression in the manuscript.

2. The authors mentioned that they downloaded 1358 experiments with raw sequence from 14 projects? Did the authors mean 1358 samples? Did the authors check the detailed sample information of the 14 projects, and confirm the samples are cell-free DNA samples from the blood?

In Figure 1, it mentions 1012 experiments with raw sequence from 14 projects were downloaded from SRA. Please double check to make the statement consistent across the manuscript.

3. A few questions about the survival analysis:

1) It mentioned 10,967 samples in the Methods section, and 10,953 patients in the results section. Please check the numbers and make it consistent.

2) Are these samples used for survival analysis also cell-free DNA samples from the blood? Please clarify.

3) Please have a rationale explaining why not use the same database (used for variant calling) for survival analysis.

4) Are the mutation of these genes correlated with changed gene expression? 

Some proofreading is needed to improve the overall quality of the manuscript.

Author Response

(The authors gave the same response as above.)

Round 2

Reviewer 1 Report

Authors did not properly explain comments no. 3.  The ctDNA copy number in tumors is often reduced, but does cfDNA still increase? And if so, what explanation can be given for this mechanism?

Author Response

We appreciate the reviewer's input on this matter. I assume the reviewer is referring to the amplification in Figure 5. Our study was supported by data from fourteen cfDNA projects. However, we lack the necessary high-quality genetic and survival data for these 14 projects, such as copy number variation. Therefore, we mapped our top 116 candidate genes to the TCGA mutational data, which includes the gene amplification data, so that we could utilise information regarding patient survival rates for the survival analysis. Simply put, these amplification observations are based solely on the public data from the TCGA pan-cancer study and not the cfDNA datasets that we employed.

We did not observe a decrease in the number of ctDNA copies or an increase in cfDNA. Our copy number variation analysis is based on publicly available information. Therefore, there was no mechanism-level explanation for the relationship between ctDNA decrease and cfDNA increase.